# Bridging pico-to-nanonewtons with a ratiometric force probe for monitoring nanoscale polymer physics before damage

Ryota Kotani[1], Soichi Yokoyama [1], Shunpei Nobusue [2], Shigehiro Yamaguchi [3], Atsuhiro Osuka[1], Hiroshi Yabu [4✉] & Shohei Saito [1,5✉]

Understanding the transmission of nanoscale forces in the pico-to-nanonewton range is important in polymer physics. While physical approaches have limitations in analyzing the local force distribution in condensed environments, chemical analysis using force probes is promising. However, there are stringent requirements for probing the local forces generated before structural damage. The magnitude of those forces corresponds to the range below covalent bond scission (from 200 pN to several nN) and above thermal fluctuation (several pN). Here, we report a conformationally flexible dual-fluorescence force probe with a theoretically estimated threshold of approximately 100 pN. This probe enables ratiometric analysis of the distribution of local forces in a stretched polymer chain network. Without changing the intrinsic properties of the polymer, the force distribution was reversibly monitored in real time. Chemical control of the probe location demonstrated that the local stress concentration is twice as biased at crosslinkers than at main chains, particularly in a strain-hardening region. Due to the high sensitivity, the percentage of the stressed force probes was estimated to be more than 1000 times higher than the activation rate of a conventional mechanophore.

[1] Graduate School of Science, Kyoto University, Kyoto 606-8502, Japan. [2] Institute of Advanced Energy, Kyoto University, Uji 611-0011, Japan. [3] Graduate School of Science, Nagoya University, Nagoya 464-8602, Japan. [4] WPI-Advanced Institute for Materials Research (AIMR), Tohoku University, Sendai 980-8577, Japan. [5] PRESTO, Japan Science and Technology Agency, Kyoto 606-8502, Japan. ✉email: hiroshi.yabu.d5@tohoku.ac.jp; saito.shohei.4c@kyoto-u.ac.jp

Understanding the transmission of pico-to-nanonewton forces in complex hierarchical structures is a primary goal in polymer physics[1–3] and mechanobiology[4,5]. Physical approaches such as optical/magnetic tweezers and atomic force microscopy (AFM) are widely used to evaluate piconewton (pN) force, but it is difficult to visualize the distribution of forces in condensed matter or molecular crowding environments with these techniques (Fig. 1a)[6–8]. While there are several instruments, such as surface force apparatuses (SFAs), rheometers, and tensile testing machines, that can analyze physical forces transmitted in meso-to-macroscopic structures, methods to directly quantify the distribution of forces at the molecular level are still being pursued. A promising alternative is chemical doping of mechanoresponsive molecules as a force probe[9–14]. These chromophores for materials and biological systems have been independently developed over the last few decades because the force magnitudes of interest are different in these fields. In the field of mechanobiology, Förster resonance energy transfer (FRET) dyads have been developed to analyze forces ranging from several to tens of pN[15–18]. On the other hand, various mechanophores respond to forces with higher thresholds involving internal covalent bond scission, which have produced a number of stress-responsive materials[19–32]. In particular, spiropyrans bearing different substituents have been reported to isomerize on 100 ms timescales under forces of 200–400 pN[33,34], while density functional theory (DFT) calculations of most mechanophores have provided force thresholds on

the order of a few nanonewtons[35,36]. Reversible responses have also been obtained in several mechanophores, even after covalent bond scission[37–40]. Recently, spiropyrans have also been used as a force probe to address important questions in polymer physics, including those related to load transmission and distribution at the single-chain (segmental) level[9–14]. However, there are still stringent requirements for the design of force probes that can quantitatively monitor local stress concentrations before the polymer chain network is structurally damaged but without changing the intrinsic properties of the polymer by chemical doping. Although sacrificial bond scission of polymer materials has been monitored using turn-on mechanophores[9–11], fully understanding the distribution of nanoscale forces in entangled polymer chains is still challenging, and its importance becomes more obvious in the rational design of uniquely tough materials[41–49]. The target force range for this purpose is ~10–100 pN, which is below that for covalent bond scission (200 pN to several nN) and beyond that for spontaneous thermal fluctuation at room temperature ($k_BT = 4.1$ pN nm)[16,50] (Fig. 1b). Here, we demonstrate that a conformationally flexible flapping force probe enables this study by ratiometric analysis of its bright dual fluorescence (FL), in which the potential energy profiles in the ground state ($S_0$) and the lowest singlet excited state ($S_1$) are suitably designed. This high brightness allows minimal chemical doping to preserve the intrinsic polymer properties. Controlling the chemical location of the probe in crosslinked polymers and

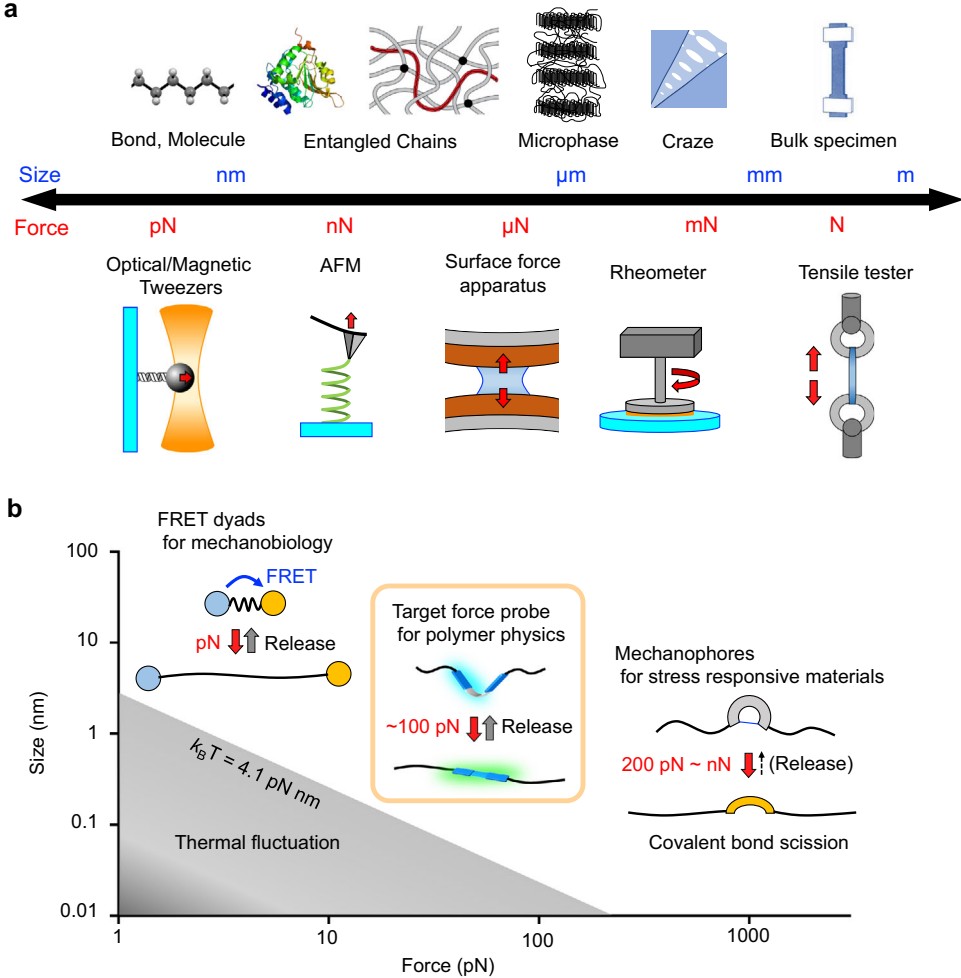

**Fig. 1 Material sizes and forces correlated in common measurements. a** Hierarchical structures of materials and physical measurement methods of mechanical forces at different scales. **b** Target force range for the nanoscale study of polymer physics.

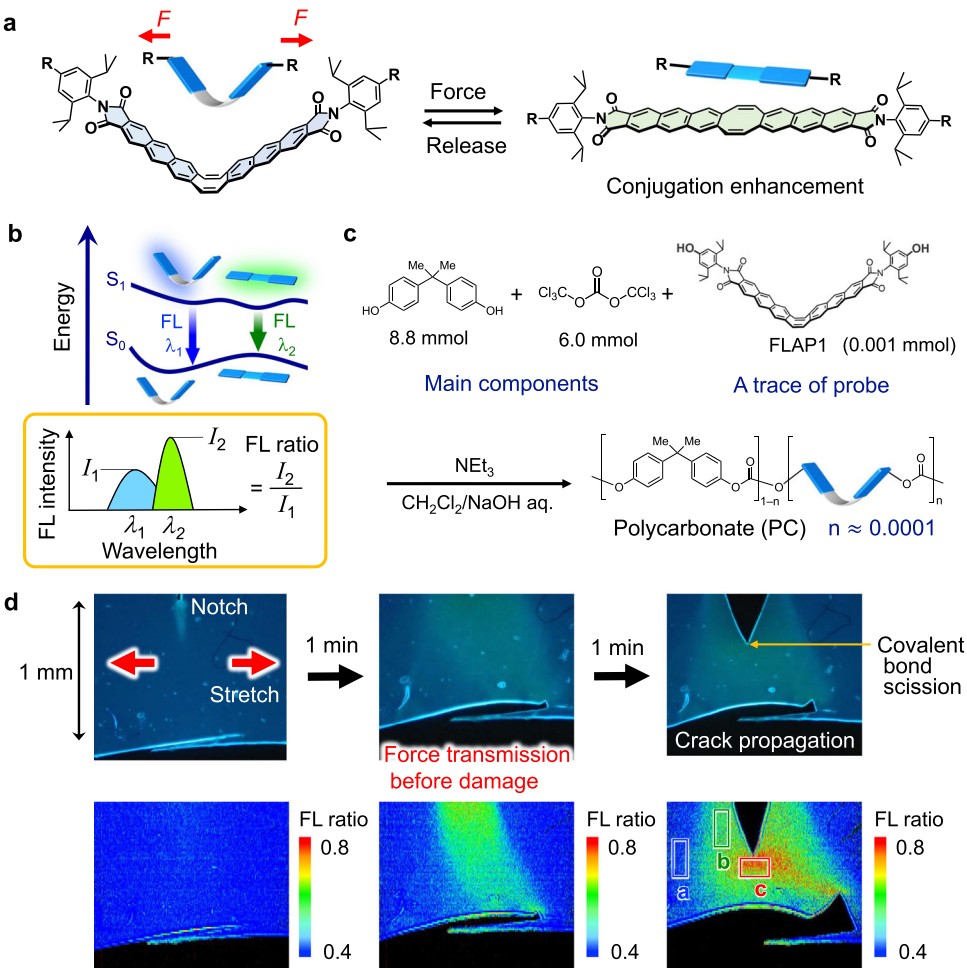

**Fig. 2 Conformationally flexible force probe and the application to ratiometric fluorescence imaging. a** Flapping molecular force probe (FLAP) that enables **b** quantitative ratiometric analysis based on dual fluorescence (FL). **c** Preparation of polycarbonate (PC) from bisphenol A and triphosgene chemically doped with trace amounts of **FLAP1** (0.05 wt%). **d** FL microscopy images of the notched PC specimen under 365-nm UV irradiation. Force transmission induced before crack propagation by tensile testing (strain rate of $2.0 \times 10^{-3}$ s$^{-1}$) (top). Corresponding two-dimensional ratiometric FL images (FL$_{525}$/FL$_{470}$) obtained by a hyperspectral camera (bottom). In the last image, an unstretched region (A), a backward region of the crack (B), and the crack tip front (C) are indicated.

observing the dual emission from the stretched polymers in real time provided new quantitative insights into biased nanoscale stress concentrations.

## Results and discussion
### Dual-fluorescence flapping molecular probe
*Dual-fluorescence flapping molecular probe.* A flapping molecule (FLAP), composed of two rigid anthraceneimide wings fused with a conformationally flexible eight-membered ring (cyclooctatetraene, COT), shows dual FL properties (Fig. 2a, b)[51]. In S$_0$, a bent form is most stable, while a planar form has higher energy due to the strain effect of the flat COT ring[52]. On the other hand, FLAP has two energy minima in S$_1$ with bent and planar geometries. Emissions from these minima are both allowed, exhibiting blue FL at ~460 nm and green FL at ~520 nm. Since the energy barrier is low and the planar geometry is slightly more stable in S$_1$, FLAP changes its conformation from bent to planar on a subnanosecond timescale after ultraviolet (UV) excitation in solution phase[53], emitting predominantly green FL with a high FL quantum yield ($\Phi_F \approx 0.3$). The population of excited species at these S$_1$ minima is largely dependent on the local viscosity in solution, while the polarity dependence can be ignored due to the non-charge-transfer character. Whereas the viscosity-probing function

is active in the presence of solvent molecules, the planarization dynamics across the barrier in S$_1$ is suppressed in a polymer matrix, such that blue FL is mainly observed ($\Phi_F \approx 0.3$). On the other hand, planarization in S$_0$ can be mechanically induced by an external force, as we previously demonstrated the mechanophore function in a crystal phase transition of FLAP[54]. Among conformationally flexible mechanophores that can show detectable signals without covalent bond scission[55-62], FLAP is unique in its dual FL properties, which enables ratiometric FL analysis (Fig. 2b). Although a variety of single-component dual-fluorescence chromophores have been reported in photochemistry[63,64], force probe application has not been reported. By ratiometric FL analysis, quantitative evaluation can be realized even when the molecular force probes are distributed heterogeneously and the concentration of the probe changes by polymer deformation.

It is also meaningful to compare the working mechanism of the flapping force probe with that of the reported rigidochromic probes (categorized into fluorogenic molecular rotors), which can provide important microscopic insights into the friction force on a real contact area[65,66]. The similarity between the two probes can be seen in the mechanical control of the distribution of different excited-state species. On the other hand, the difference is the direction of the mechanically induced dynamics; namely, in the rigidochromic probe, the excited-state dynamics from the nearly

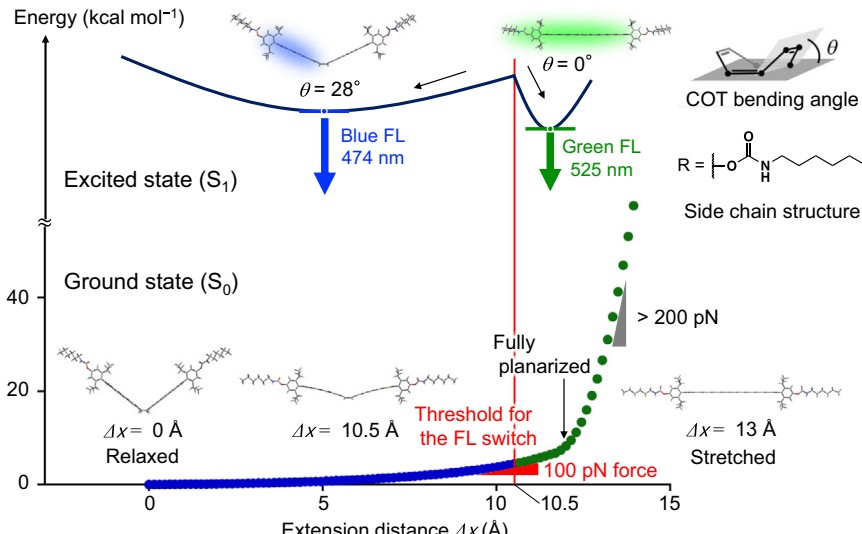

**Fig. 3 DFT-calculated energy profile of a polymer-chain substructure in the vicinity of FLAP (R = *N*-hexylcarbamate in Fig. 2a).** Constrained geometry optimization was performed in $S_0$ at the PBE0/6-31G(d) level (bottom), in which the distance between terminal carbon atoms was fixed. From the most relaxed geometry, the relative potential energy was scanned as a function of extension, $\Delta x$, by 0.15 Å. Excited-state geometry optimization in $S_1$ at the TD-PBE0/6-31G(d) level was also performed (top) for each geometry obtained in $S_0$, where the extension was fixed at the same length. The presence of two relaxed conformers of the FLAP skeleton was indicated in $S_1$, whose COT bending angles were 28° (bent) and 0° (planar). The conformational coordinates along the abscissa in $S_1$ (top) are different from those of $S_0$ (bottom) in this figure.

Franck-Condon emissive geometry to another nonemissive geometry is suppressed in the confined space under mechanical compression. In contrast, the flapping force probe is initially confined with the bent form in a narrow free volume of the polymer materials[67], but the population of the planarized species increases due to mechanical tension in the stretched polymer chain network (vide infra).

### Force threshold for fluorescence switching before mechanical bond scission

*Force threshold for fluorescence switching before mechanical bond scission.* Flapping molecules bearing OH groups were prepared for covalent doping in polycarbonate (PC) and polyurethane (PU). Terminal bulky substituents were introduced to suppress aggregation of the FLAP probes[68]. Due to the high brightness, the doping ratio could be minimized to <$10^{-4}$ equivalent to that of other monomer components. The original mechanical properties of the host polymers were preserved even after covalent doping (Supplementary Fig. 20 for PC and Supplementary Fig. 27 for PU), which is an essential requisite for a force probe. The average distance between the doped FLAPs was estimated to be ≈20 nm. Since the distance is much farther than that for FRET (<10 nm), the FRET-induced FL perturbation can be ignored. Negative control experiments by physical (noncovalent) doping of FLAP confirmed almost no mechanical response in terms of FL (Supplementary Fig. 38). In polymeric environments with chemical (covalent) doping, the sensitive mechanical response of this probe was confirmed. A linear PC sample prepared from the components in Fig. 2c showed glassy (nonelastomeric) behavior, with a glass transition temperature $T_g$ of 153 °C (Supplementary Fig. 19). When the notched PC film was slowly stretched under an inverted microscope equipped with a small tensile tester and a hyperspectral camera, the rapid growth of the stressed area was visualized on a micro- to millimeter scale based on the resulting FL, which changed from a bluish to greenish color under UV light (Supplementary Movie 1). Two-dimensional imaging as a function of the FL ratio at 525- and 470-nm intensities ($FL_{525}/FL_{470}$) clarified the situation. The spectral response started much earlier

than crack propagation from the notch (Fig. 2d). This result strongly suggested that the conformational planarization of FLAP can be specifically induced prior to covalent bond scission of the stressed polymer chains. In addition, compared with that in the crack tip front (region C), the FL ratio in a backward region of the crack (region B) decreased. This observation indicates that the FLAP force probe can trace the spatially transmitted relaxation process, while mechanically irreversible chromophores have a limitation in tracing relaxation after activation[69].

To estimate the threshold of the FL switch at the single-molecule level, (time-dependent) DFT calculations were performed in $S_0$ and $S_1$ on a polymer substructure near FLAP (Fig. 3). The energy profile in $S_0$ was obtained by plotting the relative potential energy of the optimized geometries with a fixed distance between terminal carbon atoms. At the most relaxed structure with a minimum energy, the corresponding distance was 40.4 Å (where extension = 0 Å), and the COT bending angle was 40.2°. Scanning the extension until 15 Å indicated that, in the early stage, the planarization of FLAP occurs specifically. The planarization completed by 12.0-Å extension is an energetically uphill process, but it requires only 6.7 kcal/mol. The steepness of the early-stage slope, which is the required force for conformational planarization, is several tens of pN. Further extension results in a rapid energy increase over a 200-pN slope. Eventually, the side chain is broken without bond scission in the FLAP skeleton. To determine the threshold for FL switching, the corresponding $S_1$ energy profile must also be considered because a small conformational relaxation in $S_1$ should be expected, even in the polymeric environment that suppresses a large structural change in principle[67]. Therefore, additional optimizations in $S_1$ were conducted with the same extension value at each geometry as obtained in $S_0$. As a result, $S_1$ optimization from half-stretched structures below and above the 10.5-Å extension converged into two specific conformations of the FLAP skeleton, in which the COT bending angles were 28° (bent) and 0° (planar), respectively. The calculation result is consistent with the experimental results. The FL spectra of the stretched FLAP-doped PUs (see Fig. 4) showed a spectral change in only the relative intensity of the two FL bands at 474 and 525 nm, without a gradual redshift in

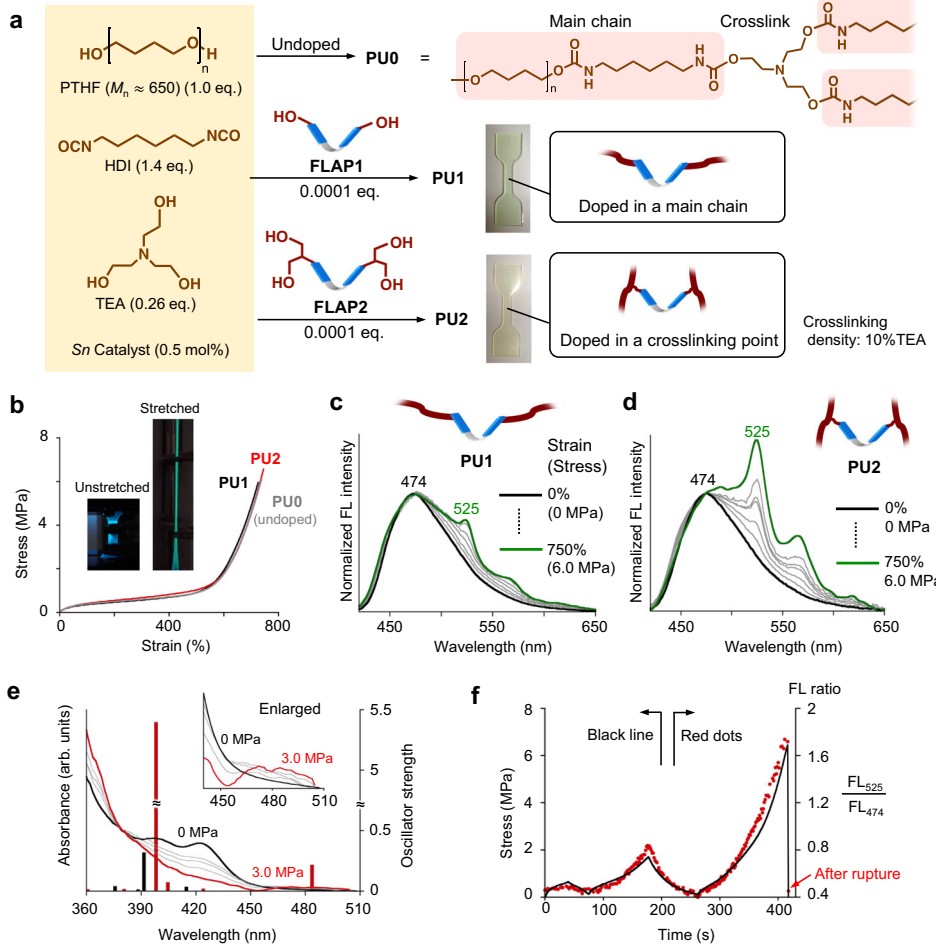

**Fig. 4 Controlled chemical doping of the force probe in crosslinked polyurethanes and the mechanical/optical properties under the uniaxial stretching conditions. a** Doping of **FLAP1** and **FLAP2** in crosslinked polyurethanes, **PU1** and **PU2**, respectively. PTHF: poly(tetrahydrofuran); HDI: hexamethylenediisocyanate; TEA: triethanolamine; Sn catalyst: dibutyltin dilaurate. The crosslinking density can be tuned by the molar ratio of TEA to the other monomer components. **b** Engineering stress–strain curves of rubbery polyurethanes, **PU1**, **PU2** and undoped **PU0** with a crosslinking density of 10% TEA. Inset images: unstretched and stretched fluorescent specimens of **PU2** under UV light. **c**, **d** FL spectral changes during the tensile testing of **PU1** and **PU2**. FL intensity was normalized at 474 nm. Excitation wavelength: 365 nm. **e** An absorption spectral change of **PU2** by stretching the specimen. Oscillator strengths were calculated at the unstretched and fully stretched geometries (Supplementary Tables 12 and 13) at the TD PBE0/6-31 + G(d) level. **f** Real-time and reversible FL response during the loading–unloading cycle of **PU2**. Time-dependent stress (black line) and FL ratio plots (red dots).

wavelength. The observed unshifted FL bands indicate that the population of the excited species basically accumulates at the two $S_1$ minima. Since the $S_1$ relaxation of the FLAP skeleton proceeds on an ultrafast timescale[53], polymer chains around the FLAP force probe cannot largely move at this instant. Based on these results, FL switching of the single FLAP molecule in the polymer substructure can be expected around the 10.5-Å extension, at which the threshold energy is calculated to be 4.5 kcal/mol with the required force of ~100 pN.

### Controlling the chemical location of the force probe in crosslinked polyurethanes

*Controlling the chemical location of the force probe in crosslinked polyurethanes.* By controlling the chemical location of the FLAP force probe, we gained new quantitative insights into the biased local stress concentration of crosslinked PUs. Two different PUs, **PU1** and **PU2**, were synthesized by chemical doping with **FLAP1** and **FLAP2**, respectively (Fig. 4a). The photophysical properties of **FLAP1** and **FLAP2** were identical in solution. Due to the different numbers and positions of the reactive OH groups, **FLAP1** is incorporated into PU main chains, while **FLAP2** is incorporated into crosslinking points. Except for the difference

in the probe locations, the following aspects of these PUs are all the same: stoichiometric amounts of monomer components, polymerization protocols using a tin catalyst, and preparation methods for dumbbell-shaped test specimens. Indeed, **PU1** and **PU2** as well as undoped PU (**PU0**) reproducibly showed almost identical mechanical and thermal properties (Fig. 4b and Supplementary Figs. 25–27). As shown Fig. 4, the crosslinking density, namely, the molar fraction of triethanolamine (TEA) to the sum of the monomers, was set to 0.10, while the molar ratio of **FLAP1** and **FLAP2** was small enough to be ignored, that is, 0.0001 equivalent to the poly(tetrahydrofuran) (PTHF) monomer (0.02 wt% to the obtained PUs). The stress–strain curve of the PU specimens indicated highly elastomeric properties (Fig. 4b). Typical rubbery deformations with Young's modulus values of 1.0–1.1 MPa were confirmed for these PUs. The yield point was not clearly observed for the rubbery PU. After a typical plateau-like region at ~1 MPa, strain-hardening behavior started at ~500% strain, and rupture eventually occurred at ~750% strain at ≈6 MPa. It is worth noting that the rubbery PUs were gradually transformed into semicrystalline PUs within several days (Supplementary Figs. 39–41); however, we focused on rubbery PUs in this study.

**a**  Crosslinking density dependence

**d**  Estimation of the stressed probe (%)

$$\Delta S = n_p \varepsilon_p \Phi_p$$

$$S_0 = n_v \varepsilon_v \Phi_v$$

Stressed probe (%)

$$= \frac{n_p}{n_v + n_p} \times 100$$

**b**  FL ratiometry against STRAIN

**c**  FL ratiometry against STRESS

**Fig. 5 Quantitative evaluation of the biased local stress concentration in crosslinked polyurethanes. a** Stress–strain curves of undoped **PU0** with different crosslinking densities. Almost the same curves were obtained for **PU1** and **PU2** at each crosslinking density (Supplementary Fig. 27). **b**, **c** FL ratio (filled) and percentages of the stressed FLAP probe (empty) as a function of macroscopic strain (b) and stress (c) based on tensile testing. Error bars in the stressed probe (%) indicate the standard deviation obtained from three tensile tests. **d** Estimation of the stressed probe (%) over the threshold force of approximately 100 pN based on the observed FL spectrum. *n*: number of the FLAP molecules, *ε*: molar absorption coefficient at the excitation wavelength (365 nm), and *Φ*: FL quantum yield, where the subscripts of "v" and "p" mean V-shaped (bent) and planar conformers, respectively. See the details in Supplementary Fig. 37.

## Fluorescence spectroscopy simultaneously conducted with tensile testing

*Fluorescence spectroscopy simultaneously conducted with tensile testing.* During mechanical testing, the FL spectra of the PU specimens were measured in real time (Fig. 4c, d). Under continuous UV irradiation (365 nm, ≈100 mW cm$^{-2}$), the FL emission was collected by an optical fiber connected to a multichannel photodetector. Before stretching, both **PU1** and **PU2** showed the same blue FL band with a single peak at 474 nm, corresponding to the emission from the bent FLAP conformation in S$_1$. As the stress increased, a green FL band at 525 nm appeared along with vibronic subbands at 565 and 618 nm emitted from the planarized conformation. On the other hand, the blue FL band did not disappear in the late stage of stretching, meaning that considerable numbers of force probes were still relaxed even just before the specimens ruptured. Namely, the macroscopic stress was not ideally distributed over the whole polymer network. The nanoscale stress concentration cannot be discussed by using "turn-on" mechanophores that show no characteristic signal in their unactivated forms. More importantly, the relative degree of green FL enhancement was remarkably larger in **PU2** than in **PU1**, clearly indicating biased nanoscale stress concentrations at the crosslinkers compared with the PU main chains. A stretch-induced absorption spectral change was also confirmed for **PU2** (Fig. 4e). The appearance of a detectable absorption band in the long-wavelength region reaching 500 nm provided strong

evidence for the compulsory conformational change of FLAP in the ground state. This weak absorption band was assigned to a partially allowed excitation in the visible range (Supplementary Table 13) for the effectively π-conjugated system of the fully planarized FLAP conformation in $S_0$. A loading–unloading cycle test was also carried out to confirm the repeated and real-time response of the mechanically reversible force probe (Fig. 4f and Supplementary Fig. 36). The FL ratio excellently traced the time-dependent stress macroscopically loaded on the PU specimen. After rupture, the FL ratio immediately returned to the extent of the unstretched initial phase, which confirmed the reversible response of the FLAP force probe and the negligible influence of photodegradation.

## Quantifying biased nanoscale stress concentrations by ratiometric fluorescence analysis

*Quantifying biased nanoscale stress concentrations by ratiometric fluorescence analysis.* The biased stress concentration at crosslinking points was demonstrated in every PU with different crosslinking densities of 8.7%, 10%, and 13% TEA (Fig. 5). The engineering stress–strain curves of these PUs are shown in Fig. 5a. The rupture stress and rupture strain decreased with increasing crosslinking density. The strain-hardening region of the highly crosslinked PU (13% TEA) started earlier at ≈400% strain, while that of the other PUs started at ≈500% strain. The FL ratios ($FL_{525}/FL_{474}$) were plotted as a function of macroscopic strain and stress (Fig. 5b, c), respectively. Here, we obtained an important insight. For all **PU2** specimens containing FLAP at crosslinking points, the FL ratio remarkably increased in the strain-hardening region. This behavior is in sharp contrast to the moderately increased FL ratio in the **PU1** specimens with FLAP at the main chains. In the plateau-like region of the stress–strain curve (observed below 1 MPa stress), the FL ratios were not largely different between **PU1** and **PU2**, indicating that the local stress concentrations were observed to the same degree at the main chains and crosslinkers. However, the biased stress concentration at the crosslinkers became pronounced during the strain-hardening process above 1 MPa stress. With decreasing crosslinking density, the difference in the FL ratios between **PU1** and **PU2** at the rupture point increased. On the other hand, when we focused on the specific strain level in a strain-hardening region (for example, 600% strain), larger macroscopic stress was generated for more highly crosslinked polymers, and accordingly, the corresponding FL ratios were relatively larger at the same strain level. Based on spectral separation analysis of the dual FL signals, we estimated the corresponding percentage of the number of stressed FLAP probes over the threshold for FL switching (Fig. 5d and Supplementary Fig. 37), where the efficiencies of UV absorption and FL emission of the bent and planar conformations were theoretically considered. At a 10% TEA crosslinking density, the estimated percentage of the stressed probes at the rupture point reached $18 \pm 2\%$ (**PU2**) among the crosslinking FLAPs, whereas that in the main chains was $\sim 9 \pm 1\%$ (**PU1**). Note that the low threshold of the FLAP force probe (theoretically predicted to be ≈100 pN) made the stressed probe percentage more than 1000 times larger than that of a reported mechanophore (diarylbibenzofuranone) that requires covalent bond dissociations[70], demonstrating the suitability of FLAP for polymer physics studies.

## Discussion

A ratiometric force probe with the dual fluorescence properties has been developed based on the design of flapping molecules (FLAP) bearing the rigid wings and the flexible joint. The stress-induced fluorescence spectral response, originating from the reversible conformational planarization, allows us to monitor stress concentration at the molecular level in deformed polymer chain networks prior to the structural damage by covalent bond scission. DFT calculations estimated the force threshold of the fluorescence switch at the single-molecule level to be ~100 pN, which corresponds to the force range below covalent bond scission (from 200 pN to several nN) and above thermal fluctuation (several pN). Chemically controlled doping of the force probe has led to new insights into the rheology of the crosslinked polymers. Namely, in a strain-hardening region, the local stress concentration is almost twice as biased at crosslinkers than at main chains. The ratiometric force probe has great potential to explore many other fundamental insights into the polymer physics of elastic materials.

## Methods

**Synthesis and purification**. All reagents and solvents were of commercial grade and were used without further purification unless noted. Tetrahydrofuran (THF) was dried using a glass contour solvent purification system. Superdehydrated dimethylformamide (DMF) was purchased from Wako Chemicals. Poly(tetrahydrofuran) ($M_n \sim 650$) was dried under vacuum at 70 °C for 2 h before use. Thin-layer chromatography (TLC) was performed with silica gel 60 $F_{254}$ (Merck). Column chromatography was performed using Wako gel C-300.

**Measurements**. $^1H$ and $^{13}C$ nuclear magnetic resonance (NMR) spectra were recorded on a JEOL ECA-600 (600 MHz for $^1H$ and 151 MHz for $^{13}C$ NMR) spectrometer. Chemical shifts were expressed as δ in ppm relative to the internal standards CHCl$_3$ ($\delta = 7.26$ ppm for $^1H$ and $\delta = 77.16$ ppm for $^{13}C$) and DMSO ($\delta = 2.50$ ppm for $^1H$ and $\delta = 39.52$ ppm for $^{13}C$). High-resolution atmospheric-pressure chemical ionization time-of-flight mass spectrometry (HR-APCI-TOF-MS) was performed on a BRUKER micrOTOF system in positive mode. Ultraviolet (UV)–visible absorption spectra were recorded on a Shimadzu UV-3600 spectrometer. Fluorescence (FL) spectra were recorded on a JASCO FP-8500 spectrofluorometer. Absolute FL quantum yields were determined on HAMAMATSU C9920-02S system. The FL lifetime was recorded on a Hamamatsu Photonics Quantaurus-Tau C11367 spectrometer. Tensile tests were carried out using a SHIMADZU AUTOGRAPH AGS-X tester with a 1-kN load cell at the crosshead. Differential scanning calorimetry (DSC) was performed on a Hitachi High-Tech TA7000 system. Rheological measurements were conducted on an Anton Paar MCR702 rheometer. Real-time FL measurements of PU specimens were performed using an Otsuka Electronics MCPD-6800 multichannel photodetector. Crossed Nicols images were obtained by a Leica DM2500 P optical microscope equipped with a Linkam LTS420E temperature control system. Photographs and movies of specimens were taken using an OLYMPUS Tough TG-6 digital camera.

**Fluorescence microscopy imaging of the stretched polycarbonate film**. This experiment was conducted by the combinational use of an inverted microscope (OLYMPUS IX83 inverted research microscope equipped with a U-HGLGPS mercury lamp), a tensile testing machine (AcroEdge OZ911 tensile testing machine) and a hyperspectral camera (EBAJAPAN NH-8 hyperspectral camera). A polycarbonate (PC) specimen (width ≈ 1 mm and thickness ≈ 0.03 mm) with a small notch was used for tensile testing. This specimen was held with an initial distance of 25 mm and extended from both sides at a speed of 0.05 mm s$^{-1}$. Because of symmetrical stretching, the growing notch fits within the field of view of the microscope. At the same time, FL spectra at each pixel were collected with the hyperspectral camera. The data were acquired with an exposure time of 500 ms at a wavelength interval of 5 nm. FL optical micrographs (Fig. 2d, top) were taken by a digital camera (OLYMPUS TG-6), while the mapping images of the FL ratio (Fig. 2d, bottom) were obtained by analytical software (EBAJAPAN HSAnalyzer) based on the collected FL spectra.

## Data availability

All data are available in the main text or the supplementary information.

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

## Acknowledgements

We thank Dr. Hiroya Abe and Dr. Yuta Saito (Tohoku Univ.) for their help in the early stage of the preliminary study. JST PRESTO (FRONTIER) and JST FOREST, Grant numbers JPMJPR16P6 and JPMJFR201L; MEXT/JSPS KAKENHI, Grant Numbers JP21H01917, JP21H05482, JP18H01952, JP20H04625, JP19KK0357, JP18H05482, and JP18J22477; Inoue Foundation for Science; Toray Science Foundation.

## Author contributions

H.Y. and S.S. conceived the concept. R.K., H.Y., and S.S. designed the experiments. R.K., S. Yokoyama, and S.N. performed the experiments. R.K. and S.S. conducted the quantum calculations. R.K. and S.S. analyzed the data. R.K. and S.S. wrote the manuscript, and S. Yamaguchi, A.O., and H.Y. proofread it.

## Competing interests

The authors declare no competing interests.
