## [Peer Review File · Nature Communications]

REVIEWER COMMENTS

Reviewer #1 (Remarks to the Author):

The authors report a mechanochromic molecule that allows estimates of single-chain (segmental) forces in bulk mechanically-loaded polymer samples by ratiometric fluorescent measurements.

The distribution of single-chain (segmental) forces in bulk polymers under mechanochemical load is an important, unsolved problem in polymer physics. Existing experimental or computational approaches suffer from poor temporaspatial resolution (usually averaging over 100 μm^2 areas and >10 s time), force resolution (usually <100 pN, 300 – 500 pN and >500 pN) and/or poor reversibility. In this context, the problem the authors are trying to solve is highly topical and a proper solution is likely to be quite impactful. The molecule is cleverly designed and the simple conformational transition underlying the effect ensures reversibility and compatibility with a range of environments. The measurements are competently performed and the demonstration of potential applications of the phenomenon, on example of quantifying the distribution of load between cross-links and segments is commendable.

My only technical complaint is the authors' seeming overconfidence in their estimate of the force required to change the fluorescence ratio. Without any data on the dynamics of conformational relaxation in S1, the computational results provided are insufficient to determine what role, if any, S0 geometry plays in determining the relative intensities of the two emission bands. Consequently, the 100 pN threshold should be viewed as very approximate, and may in practice be as low 20 pN, but probably not >150 pN, based on the plausible geometry of the S1 TS separating the two S1 minima. Because doing the required calculations properly is very demanding, the authors should acknowledge the guestimate nature of their numbers and tone down the promotion of the 100 pN value.

My main overall concern is the quality of presentation, including the analysis of the data in p. 8-9. First, the paper in the current format doesn't reflect the state of the knowledge of the field of polymer mechanochemistry and thus fails to place the findings in the proper context. The background references are either outdated or do not support the statements that the authors appear to think they do. Contemporary polymer mechanochemistry is such a diverse and rapidly developing field that citing a few original papers is unlikely to represent it correctly. Instead, several recent reviews would have been more appropriate; yet the two reviews the paper cites (refs. 14 and 28) are badly outdated and do not provide the necessary background for a non-expert to appreciate the importance of the reported findings.

It's simply incorrect to claim that mechanochemical reactions involving rearrangement of covalent bonds require nN-level forces. Dozens of spiropyrans have been reported that isomerize (often reversibly) on 100 ms-timescales under force of 200 – 400 pN. Several of those have been used to

address important questions in polymer physics, including those related to load transmission and distribution at single-chain (segmental) level. None of this work is cited by the authors.

It's also misleading to reference computational papers (e.g, refs 9,10) in support of statements about force required to accelerate covalent bond rearrangements. With few notable exceptions, quantum-chemical calculations yield very poor predictions of such forces. Instead, single-molecule force spectroscopy is the main technique to measure mechanochemical kinetics (i.e., reaction rates as a function of the force acting on the reactant), although model studies have been used as well.

The paper's prose and organization are poor; and incorrect word usage, idiosyncratic sentence structure, and paragraphs that contain multiple ideas all distract from the substance of the work. Combined with deficiencies described above they make me wonder if the current version of the paper would indeed attract sufficiently broad attention.

Reviewer #2 (Remarks to the Author):

The authors reported an interesting cyclooctatetraene (COT) based conformational mechanophore, which planarizes under relatively weak forces to induce a change in fluorescence. The force sensor is based a FLAP molecule design that Dr. Saito reported a few years ago, and I have been looking forward to its application in stress-sensing since then. I'm glad to see the authors have done such a nice and comprehensive job to realize it.

The FLAP molecule was introduced into polycarbonate and polyurethane to map the force distribution in solid samples. Although using mechanophore for force sensing has been demonstrate, the FLAP sensor has several advantages. First, they are much more sensitive than covalent mechanophores. By DFT calculations, the author showed the planarization at around 100 pN, even before the polymer C-C bond is elongated (~200 pN). The high sensitivity could enable early detection of regions experiencing stress before covalent bond scission. This advantage was demonstrated in the paper by studying a notched PC sample. Another advantage of this system is the reversibility, which ensures multiple cycles of testing without deterioration of the materials. The authors also demonstrated that incorporating the mechanophore in crosslinks rather than polymer main chains, the sensing sensitivity can be significantly enhanced. This is a nicely written manuscript.

Reviewer #3 (Remarks to the Author):

This is an interesting paper describing the working of a new type of 'flapping' force probe that fills a hole in the range of forces that can be measured with conventional techniques. The results are enticing and important, and I think this paper merits to be published in Nature Comms. I do have a number of remarks that need to be addressed before the paper can be published.

1. The English is bad, so bad that at some points it makes the paper hard to understand. Please have a native speaker correct this.

2. It is not completely clear to me how these new flapping probes compare to 'ordinary' rigidochromic molecules that have a similar way of working, and have recently been used as force/contact probes also, see e.g. Molecular probes reveal deviations from Amontons' law in multi-asperity frictional contacts B Weber, T Suhina, T Junge, L Pastewka, AM Brouwer, D Bonn Nature communications 9 (1), 1-7 (2018) and Fluorescence microscopy visualization of contacts between objects

T Suhina et al. Angewandte Chemie 127 (12), 3759-3762 (2015).

3. Have the authors considered doing fluorescence lifetime rather than intensity measurements?

4. I have difficulties with the interpretation in terms of bond elongation by stress/strain. This is easy to check: the infrared frequencies of the vibrational modes of the molecules would be different. In the one experiment that has attempted this, no evidence for bond stretching was found: Nanoparticle amount, and not size, determines chain alignment and nonlinear hardening in polymer nanocomposites

HS Varol et al. Proceedings of the National Academy of Sciences 114 (16), E3170-E3177 (2017)

If these comments can be dealt with, I would gladly recommend this manuscript for publication

2021 Nov 15

Revision of the manuscript NCOMMS-21-05858A

“Bridging pico-to-nanoNewton: Ratiometric Force Probe for Nanoscale Polymer Physics Before Damage”

Ryota Kotani, Soichi Yokoyama, Shunpei Nobusue, Shigehiro Yamaguchi, Atsuhiko Osuka, Hiroshi Yabu*, Shohei Saito*

We here revised our manuscript according to the comments and suggestions from the reviewers. We apologize for this late revision due to the limited research activity by COVID-19. What we have changed is listed below. To expedite the review process, all the above changes are marked in yellow in the revised manuscript.

Shohei Saito
Associate Professor
Kyoto University

Our reply to the first reviewer's comments:

The authors report a mechanochromic molecule that allows estimates of single-chain (segmental) forces in bulk mechanically-loaded polymer samples by ratiometric fluorescent measurements.

The distribution of single-chain (segmental) forces in bulk polymers under mechanochemical load is an important, unsolved problem in polymer physics. Existing experimental or computational approaches suffer from poor temporaspatial resolution (usually averaging over 100 μm^2 areas and >10 s time), force resolution (usually <100 pN, $300 - 500$ pN and >500 pN) and/or poor reversibility. In this context, the problem the authors are trying to solve is highly topical and a proper solution is likely to be quite impactful. The molecule is cleverly designed and the simple conformational transition underlying the effect ensures reversibility and compatibility with a range of environments. The measurements are competently performed and the demonstration of potential applications of the phenomenon, on example of quantifying the distribution of load between cross-links and segments is commendable.

We are grateful for his/her positive comments based on his/her deep understanding in molecular and polymer mechanochemistry. As the reviewer mentioned, we hope that the conceptual topic of our work attracts broad interests in multidisciplinary fields. We also appreciate the many fruitful suggestions to improve the quality of the manuscript. Our point-by-point responses are summarized below.

My only technical complaint is the authors' seeming overconfidence in their estimate of the force required to change the fluorescence ratio. Without any data on the dynamics of conformational relaxation in S_1 , the computational results provided are insufficient to determine what role, if any, S_0 geometry plays in determining the relative intensities of the two emission bands. Consequently, the 100 pN threshold should be viewed as very approximate, and may in practice be as low 20 pN, but probably not >150 pN, based on the plausible geometry of the S_1 TS separating the two S_1 minima. Because doing the required calculations properly is very demanding, the authors should acknowledge the guestimate nature of their numbers and tone down the promotion of the 100 pN value.

As the reviewer pointed out, the corresponding description on the force threshold for the fluorescence switching was revised. In the revised version, the force threshold has been mentioned just as a theoretically estimated value about 100 pN. As s/he suggested, it is very difficult to estimate the accurate value of the threshold in consideration of the S_1 excited state energy profile. To experimentally determine the threshold, combinational application of single-molecule fluorescence spectroscopy and single-molecule force analysis using AFM or optical/magnetic tweezers is required, but it is out of the scope of this article.

My main overall concern is the quality of presentation, including the analysis of the data in p. 8-9. First, the paper in the current format doesn't reflect the state of the knowledge of the field of polymer mechanochemistry and thus fails to place the findings in the proper context. The background references are either outdated or do not support the statements that the authors appear to think they do. Contemporary polymer mechanochemistry is such a diverse and rapidly developing field that citing a few original papers is unlikely to represent it correctly. Instead, several recent reviews would have been more appropriate; yet the two reviews the paper cites (refs. 14 and 28) are badly outdated and do not provide the necessary background for a non-expert to appreciate the importance of the reported findings.

We have added the latest reviews and key papers up to the limitation of the reference number (70 references), which refer to the advanced polymer mechanochemistry as well as the mechanobiology and the single-molecule force spectroscopy. Regarding the analysis of the data on p. 8-9, we have discussed many times with several experts in the field of polymer rheology and we started a new collaboration for interpreting these experimental data more deeply by the aid of coarse-grained MD calculations of cross-linked polymers. After that, now we are sure that the discussion on p. 8-9 should be sound from the viewpoint of the polymer rheology. The results of the coarse-grained MD calculations will be reported in near future.

It's simply incorrect to claim that mechanochemical reactions involving rearrangement of covalent bonds require nN-level forces. Dozens of spiropyrans have been reported that isomerize (often reversibly) on 100 ms-timescales under force of 200 – 400 pN. Several of those have been used to address important questions in polymer physics, including those related to load transmission and distribution at single-chain (segmental) level. None of this work is cited by the authors.

We thank the reviewer for this important correction. We particularly added the description on the 200–400 pN force thresholds of spiropyrans bearing different substituents, determined by single molecule force spectroscopy using AFM (S. L. Craig et al. *JACS* 2015, 137, 6148; *JACS* 2018, 137, 6148). Fig.1B has been revised accordingly. The latest references of the mechanochemistry were also added for demonstrating important insights into polymer physics at the molecular level.

It's also misleading to reference computational papers (e.g, refs 9,10) in support of statements about force required to accelerate covalent bond rearrangements. With few notable exceptions, quantum-chemical calculations yield very poor predictions of such forces. Instead, single-molecule force spectroscopy is the main technique to measure mechanochemical kinetics (i.e., reaction rates as a function of the force acting on the reactant), although model studies have been used as well.

To distinguish the experimental results of SMFS from the theoretical predictions made by the CoGEF (constrained geometries simulate external force) calculations, we clearly separated these sentences and the related citations in the revised version.

The paper's prose and organization are poor; and incorrect word usage, idiosyncratic sentence structure, and paragraphs that contain multiple ideas all distract from the substance of the work. Combined with deficiencies described above they make me wonder if the current version of the paper would indeed attract sufficiently broad attention.

The entire English text has been polished and simplified according to the advice from a native speaker. We revised the structure of the sentences particularly in the abstract and introduction. This research is deeply related to molecular photochemistry, ratiometric fluorescence imaging and polymer synthesis as well as mechanochemistry. Although the description is multidisciplinary, now we hope the readability became better. Together with the comments of the other reviewers, we believe that the conceptual advances in designing the ratiometric fluorescent force probe and its application for polymer physics are worthy of publication after appropriate English modifications and proper background descriptions, including the latest reference citations.

Our reply to the second reviewer's comments:

The authors reported an interesting cyclooctatetraene (COT) based conformational mechanophore, which planarizes under relatively weak forces to induce a change in fluorescence. The force sensor is based a FLAP molecule design that Dr. Saito reported a few years ago, and I have been looking forward to its application in stress-sensing since then. I'm glad to see the authors have done such a nice and comprehensive job to realize it. The FLAP molecule was introduced into polycarbonate and polyurethane to map the force distribution in solid samples. Although using mechanophore for force sensing has been demonstrate, the FLAP sensor has several advantages. First, they are much more sensitive than covalent mechanophores. By DFT calculations, the author showed the planarization at around 100 pN, even before the polymer C-C bond is elongated (~200 pN). The high sensitivity could enable early detection of regions experiencing stress before covalent bond scission. This advantage was demonstrated in the paper by studying a notched PC sample. Another advantage of this system is the reversibility, which ensures multiple cycles of testing without deterioration of the materials. The authors also demonstrated that incorporating the mechanophore in crosslinks rather than polymer main chains, the sensing sensitivity can be significantly enhanced. This is a nicely written manuscript.

We appreciate his/her recommendation for the acceptance of our work for publication. The reviewer clearly pointed out the originalities of our work.

Our reply to the last reviewer's comments:

This is an interesting paper describing the working of a new type of 'flapping' force probe that fills a hole in the range of forces that can be measured with conventional techniques. The results are enticing and important, and I think this paper merits to be published in Nature Comms. I do have a number of remarks that need to be addressed before the paper can be published.

We are glad to receive the high evaluation on the originality of our molecular system and the idea of the ratiometric force mapping in a real time manner.

1. The English is bad, so bad that at some points it makes the paper hard to understand. Please have a native speaker correct this.

Thank you for the advice. The entire English text has been polished and simplified according to the advice from a native speaker. We revised the structure of the sentences particularly in the abstract and introduction.

2. It is not completely clear to me how these new flapping probes compare to 'ordinary' rigidochromic molecules that have a similar way of working, and have recently been used as force/contact probes also, see e.g. Molecular probes reveal deviations from Amontons' law in multi-asperity frictional contacts B Weber, T Suhina, T Junge, L Pastewka, AM Brouwer, D Bonn Nature communications 9 (1), 1-7 (2018) and Fluorescence microscopy visualization of contacts between objects. T Suhina et al. Angewandte Chemie 127 (12), 3759-3762 (2015).

As stated by the reviewer, it makes sense to compare the present work with the reported application of the rigidochromic probes, in which an important microscopic insight into the friction force on real contact area has been successfully obtained. In these reports, compression of materials onto the glass substrate surface suppresses the non-radiative motion of photo-excited species covalently bound to the glass surface, resulting in a quantitative increase in fluorescence intensity. The similarity of these works (rigidochromic probes and flapping force probes) can be seen in the mechanical control of the distribution of the different excited-state species. On the other hand, the difference would be the direction of the mechanically induced dynamics: Namely, in the rigidochromic probe, the excited-state dynamics from the nearly Franck-Condon emissive geometry to another non-emissive geometry has been suppressed in the confined space under mechanical compression. In contrast, the flapping force probe is initially confined with the bent form in the polymer materials, but the population of the planarized species increased by the mechanical tension in the stretched polymer chain network. Since the new absorption band appeared in the lowest energy region by stretching the FLAP-doped elastomers, the mechanical planarization does occur in the S_0 ground state. The increased population of the planarized emissive form would be mainly induced by the S_0 planarization rather than the redistribution of the S_1 species after the photoexcitation. In relation to this, we currently focus on the new flapping force probes that works even in organo-gels or that works even under gentle mechanical compression.

In the revised manuscript, we added the above-mentioned discussion shortly, including the references of the rigidochromic probes.

3. Have the authors considered doing fluorescence lifetime rather than intensity measurements?

We are also interested in the application of FLIM to the force mapping. There is a potential candidate for the highly photostable force probe for FLIM (*Angew. Chem. Int. Ed.* **2020**, *59*, 16430; *Bull. Chem. Soc. Jpn.* **2020**, *93*, 1102–1106). However, it would be categorized as a "turn-off" force probe in fluorescence intensity because the non-emissive planarized form would be populated by the mechanical tension.

*4. I have difficulties with the interpretation in terms of bond elongation by stress/strain. This is easy to check: the infrared frequencies of the vibrational modes of the molecules would be different. In the one experiment that has attempted this, no evidence for bond stretching was found: Nanoparticle amount, and not size, determines chain alignment and nonlinear hardening in polymer nanocomposites
HS Varol et al. Proceedings of the National Academy of Sciences 114 (16), E3170-E3177 (2017).*

We thank the reviewer for the valuable information. Although the mechanical bond elongation before the bond scission has been theoretically assumed in many literatures of mechanochemistry (for example, ref 35 and 36), the experimental result in the PNAS paper is important. Therefore, we added the short statement on the absence of the experimental evidence for the bond elongation up to date, by citing the PNAS paper.

Other non-scientific corrections made:

- Several fundings were added in Acknowledgements.

We hope that the revised manuscript has sufficiently provided appropriate answers to the suggestions. We would be glad if it would be accepted for publication.

Best Regards,

Shohei Saito
Associate Professor
Kyoto University

REVIEWERS' COMMENTS

Reviewer #1 (Remarks to the Author):

scientifically, the paper is fine and the rationalization of the results is adequately rigorous. While the prose has improved somewhat compared to the original draft, it's still very awkward and makes the paper unnecessarily hard to read. The editor is probably best placed to determine how much, if at all, the prose needs to improve before the paper is acceptable but I didn't find the current version a joy to read.

The discussion of C-C bond length upon stretching the force probe is a red-herring. The C-C bonds are the stiffest elements of the molecule and they will be least strained by applied force; the C-C bond doesn't need to elongate to experience accelerated dissociation, since (a) C-C bond distance and BDE don't generally correlate (except in introductory chemistry textbooks), nor should they based on any known law of physics and (b) acceleration of C-C bond homolysis in strained molecules proceeds not by elongation (or "weakening") of the bond but by stabilization of the TS that is longer than the reactant along the pulling axis.

Finally, COGEF has never been demonstrated to produce experimentally validated dissociation force. It's a mechanochemistry equivalent of truthiness: a construct designed to obfuscate and confuse more than clarify by attaching numbers to qualitative ideas despite the fact that these numbers have no basis in reality.

Reviewer #3 (Remarks to the Author):

The authors have done a thorough job in revising the manuscript, and I still think the reported results are enticing. So I recommend publication as it is.

Revision of the manuscript NCOMMS-21-05858A

“Bridging Pico-to-Nanonewton with a Ratiometric Force Probe for Monitoring Nanoscale Polymer Physics Before Damage”

Ryota Kotani, Soichi Yokoyama, Shunpei Nobusue, Shigehiro Yamaguchi, Atsuhiko Osuka, Hiroshi Yabu*, Shohei Saito*

Note: To avoid using the punctuation, we changed the title. Please confirm the suitability.

Our reply to the first reviewer's comments:

scientifically, the paper is fine and the rationalization of the results is adequately rigorous. While the prose has improved somewhat compared to the original draft, it's still very awkward and makes the paper unnecessary hard to read. The editor is probably best placed to determine how much, if at all, the prose needs to improve before the paper is acceptable but I didn't find the current version a joy to read.

We are grateful for his agreement with the scientific contents of our manuscript. To polish the expression of the sentences, we used the English editing service of Springer Nature for this final revision. Now we believe that the readability becomes satisfactory for native researchers.

The discussion of C-C bond length upon stretching the force probe is a red-herring. The C-C bonds are the stiffest elements of the molecule and they will be least strained by applied force; the C-C bond doesn't need to elongate to experience accelerated dissociation, since (a) C-C bond distance and BDE don't generally correlate (except in introductory chemistry textbooks), nor should they be based on any known law of physics and (b) acceleration of C-C bond homolysis in strained molecules proceeds not by elongation (or "weakening") of the bond but by stabilization of the TS that is longer than the reactant along the pulling axis.

Finally, COGEF has never been demonstrated to produce experimentally validated dissociation force. It's a mechanochemistry equivalent of truthiness: a construct designed to obfuscate and confuse more than clarify by attaching numbers to qualitative ideas despite the fact that these numbers have no basis in reality.

We thank the reviewer for this important comment. The reviewer 3 also pointed out that the "mechanical elongation" of the C–C bond has not been supported in the experiments. To avoid misleading descriptions, we removed the discussion on the calculated C–C bond length from the main text. Instead, we cited the reviewers' comments as "Note" in the Supplementary Figure 14, since this discussion is worthy to be open in the mechanochemistry community, including the researchers who utilize the COGEF calculations.

Our reply to the 3rd reviewer's comments:

The authors have done a thorough job in revising the manuscript, and I still think the reported results are enticing. So I recommend publication as it is.

We appreciate his/her recommendation for the acceptance of our work for publication. To polish the expression of the sentences, we used the English editing service of Springer Nature for this final revision. Now we believe that the readability becomes satisfactory for native researchers.

We hope that the revised manuscript has sufficiently provided appropriate answers. We would be glad if it would be accepted for publication.

Best Regards,

Shohei Saito
Associate Professor
Kyoto University